# Enhancing automated strabismus classification with limited data: Data augmentation using StyleGAN2-ADA

**Jaehan Joo**[1☯], **Sang Yoon Kim**[2☯], **Donghwan Kim**[1], **Ji-Eun Lee**[2], **Seung Min Lee**[2], **Su Youn Suh**[2], **Su-Jin Kim**[2]*, **Suk Chan Kim**[1]*

**1** Department of Electronics Engineering, Pusan National University, Busan, Republic of Korea, **2** Department of Ophthalmology, Pusan National University School of Medicine and Research Institute for Convergence of Biomedical Science and Technology Pusan National University Yangsan Hospital, Yangsan, Republic of Korea

☯ These authors contributed equally to this work.
* pearlksj@gmail.com (SJK), sckim@pusan.ac.kr (SCK)

**Data Availability Statement:** Due to patient privacy laws, the experimental results can be shared, but the remaining data used for training and testing the deep learning model cannot be disclosed to the public. Our research team and the

## Abstract

In this study, we propose a generative data augmentation technique to overcome the challenges of severely limited data when designing a deep learning-based automated strabismus diagnosis system. We implement a generative model based on the StyleGAN2-ADA model for system design and assess strabismus classification performance using two classifiers. We evaluate the capability of our proposed method against traditional data augmentation techniques and confirm a substantial enhancement in performance. Furthermore, we conduct experiments to explore the relationship between the diagnosis agreement among ophthalmologists and the generation performance of the generative model. Beyond FID, we validate the generative samples on the classifier to establish their practicality. Through these experiments, we demonstrate that the generative model-based data augmentation improves overall quantitative performance in scenarios of extreme data scarcity and effectively mitigates overfitting issues during deep learning model training.

## Introduction

Strabismus, often defined as any misalignment of the eyes, predominantly impacts children, with prevalence rates ranging from 2.0% to 5.4% [1–6]. This condition is common among children and includes variations like esotropia and exotropia [7]. Detecting and managing strabismus promptly in children is essential, given the potential for significant improvement in their long-term visual and sensorimotor results [8]. It is worth noting that strabismus stands as the primary cause of amblyopia in children [9, 10]. Furthermore, according to [11], individuals with strabismus, including children, adolescents, and adults in South Korea, demonstrated a higher risk of various mental disorders, such as developmental disorders, autism, ADHD, and obsessive-compulsive disorder(OCD) compared to the control group.

Due to the few pediatric ophthalmologists available to undertake this specialized task, numerous countries, including South Korea, amplify the importance of strabismus screening. Clinical strabismus assessments, such as the Hirschberg and Krimsky tests, are labor-intensive

data-providing institution(Pusan National University Yangsan Hospital IRB Center) have decided to restrict the public disclosure of the data during and after the research period. Access to the dataset can be requested from Pusan National University Yangsan Hospital IRB Center via +82-055-360-4722.

**Funding:** This study was supported by the Research Institute for Convergence of biomedical science and technology (30-2021-022), Pusan National University Yangsan Hospital and the 2022 BK21 FOUR Program of Pusan National University. The funders played a role in data collection and analysis, as well as preparation of the manuscript.

**Competing interests:** No authors have competing interests.

procedures requiring expert handling [12]. According to [12], even strabismus specialists can often underestimate or overestimate by a minimum of 10 prism diopters (PD) when using these techniques. Furthermore, the difficulty in achieving reliable results from children's eye examinations, such as the alternative prism cover test and extraocular muscle movement evaluation, compounds the issue. The accuracy of these results can significantly vary depending on the child's level of cooperation. Based on the facts mentioned above, we have identified the need for research to improve early detection of strabismus in children, which can lead to the prevention of associated conditions and enhance the convenience of diagnosis.

## Automated strabismus diagnosis without deep learning

Researchers have conducted imaging and computer-based studies for a more accurate and convenient diagnosis of strabismus. [13] conducted a study to evaluate the efficacy of the 3D strabismus photo analyzer(SPA) by estimating binocular alignment based on photographs. The researchers compared it to the Krimsky test and analyzed the results, finding strong agreement and improved reproducibility between the two methods. They confirmed that the 3D SPA is a reliable and convenient tool for measuring ocular deviation, making it valuable for assessing binocular alignment in clinical practice.

[14] introduced a computerized method for measuring binocular alignment using a selective wavelength filter and an infrared camera. Researchers validated the efficacy of automated image analysis by comparing it to the standard prism and alternate cover test (PCT). They conducted a prospective observational pilot study on subjects with intermittent exotropia, esotropia, and orthotropic individuals, and the results confirmed the excellent agreement between the automated image analysis with the selective wavelength filter and the PCT.

[15] conducted a study to measure the maximum angle of ocular versions using photographs of the nine cardinal positions and a modified limbus test. The image data used in the study underwent image processing using Photoshop, and the angle of the version was measured using the Image J program. The study's results revealed that the angle of maximum elevation was significantly smaller than the angle of maximum depression, and there were no correlations between the maximum angle of version and age, spherical equivalents, or axial length.

The mentioned studies propose strabismus diagnostic methods that demonstrate excellent performance through classical computer vision techniques and mathematical modeling. However, due to the multi-step process in their program usage, one cannot readily categorize these methods as fully automated end-to-end systems.

## Automated strabismus diagnosis using deep learning

Deep learning demonstrates excellent performance in the medical field and almost all areas in this era. Researchers have undertaken several studies that diagnose strabismus based on deep learning and images.

[16] proposed an automated strabismus recognition system using gaze deviation (GaDe) image data and convolutional neural networks (CNN). In their study, the researchers employed CNN models to extract feature vectors from GaDe images and used a support vector machine (SVM) classifier to determine the presence of strabismus. Experimental results demonstrated a high accuracy of 95.2% when using the VGG-S model as the feature extractor among the six representative CNN models. However, it should be noted that the proposed method in this study, similar to the previously mentioned studies, only constitutes a partial end-to-end system because it requires a separate data processing step to acquire GaDe images

from an eye tracker. Additionally, the method relies on a separate high-resolution eye tracker, which compromises its efficiency.

[17] proposed an automated Strabismus diagnosis system based on deep learning called RF-CNN. The system utilized the R-FCN model to obtain segmented eye images from human facial photographs, which were then input into a CNN model for strabismus detection. The researchers collected a dataset consisting of 5685 images specifically for this study. The dataset included 3409 training images and 2276 testing images. The experimental results showed a high accuracy of 93.89% for the proposed system, demonstrating its excellent performance in strabismus diagnosis without additional preprocessing steps. Also, [18] developed a deep learning algorithm for screening shallow horizontal strabismus using primary gaze photographs in children and compared its performance with that of ophthalmologists. In this study, a total of 7,026 images (3,829 normal and 3,197 shallow horizontal strabismus) were used to develop the deep learning algorithm. The employed deep learning model was evaluated through 5-fold cross-validation and tested on an independent validation dataset consisting of 277 images. Unlike [17], they used Faster R-CNN to crop the eye regions from the original data for analysis. The proposed deep learning system achieved a high accuracy of 95%, while the diagnostic accuracy of ophthalmologists ranged from 81% to 85%.

One of the most crucial aspects of deep learning research is dataset collection and organization. Ideally, obtaining extensive data and constructing a well-curated data set based on established collection procedures enables practical training in deep learning models. In the studies conducted by [17, 18], they collected sufficient data directly, allowing them to develop high-performance systems using simple CNN models, even without utilizing open datasets. However, if the dataset size is limited, training deep learning models becomes challenging and may lead to critical issues such as overfitting problem. Data collection is particularly challenging in the medical field compared to other research domains. Due to privacy concerns, it requires navigating complex procedures such as obtaining patient consent and obtaining institutional review board (IRB) approval.

Moreover, disease prevalence often varies based on age, gender, and race, making obtaining a representative sample data set difficult. In the case of rare diseases, constructing large-scale data sets is often impractical. Therefore, when applying artificial intelligence in the medical field, it is essential always to consider the challenge of limited data and develop strategies to address this issue.

## Challenges arising from data scarcity in deep learning

Various methods have been proposed to address the issue of data scarcity in deep learning. These methods include data augmentation [19, 20], fine-tuning [21], transfer learning [22], and regularization [23], with data augmentation and fine-tuning being the most commonly employed techniques. Data augmentation involves transforming or manipulating existing data to generate new training samples. This technique enhances the model's generalization performance by increasing the diversity of the available data. In the case of image data, common transformations include rotation, scaling, inversion, and the addition of noise. Fine-tuning refers to delicately adjusting a pre-trained model for a new task. When faced with limited data, fine-tuning utilizes the pre-trained model's initial weights and performs additional training on new data. This enables the model to leverage its existing knowledge and achieve high performance even with limited data. However, it is essential to note that while techniques such as data augmentation and fine-tuning can effectively leverage available data, they may still have limitations in situations involving extremely limited data.

As part of efforts to address these challenges, data augmentation techniques using generative adversarial networks (GANs) have been introduced [24]. GANs, initially proposed by [25], are a generative model based on deep learning. They consist of two neural networks, a generator, and a discriminator, trained competitively. The generator network generates fake data samples, while the discriminator network aims to distinguish between real and fake data. Through adversarial training, GANs learn to generate high-quality samples that are indistinguishable from real data. GANs have demonstrated remarkable performance in various applications such as image synthesis, data augmentation, and domain adaptation.

In a recent study by [26], GANs models were utilized to improve the performance of deep learning systems for strabismus diagnosis. The researchers employed the DCGAN [27] model to generate imitated data resembling strabismus samples. However, this study did not provide quantitative metrics such as fréchet inception distance (FID) or inception score (IS) to evaluate the quality of the generated samples. Additionally, there was no comparison of the results of distinguishing real strabismus cases using the generated fake data. As a result, the research findings are limited in assessing the generated samples' performance and their applicability to real-world strabismus diagnosis tasks.

## Proposed methods and contributions

In this study, we propose a method to improve strabismus classification performance using generative models in situations with limited data. We utilize StyleGAN2-ADA [28] as the generative model and compare the performance of two CNN models for strabismus classification. Additionally, we compare the results quantitatively and qualitatively when the training data set of the generative model is constructed based on the ophthalmologists' diagnostic concordance rate(ODCR). This allows us to examine the impact of expert opinions on the generation of perceptually realistic synthetic data during the training process of the generative model. The main contributions of our paper are summarized as follows.

1. The present study explores the efficacy of data augmentation using generative models in scenarios characterized by severely limited data availability.

2. We introduce a novel approach leveraging StyleGAN2-ADA as the generative model to generate synthetic strabismus data that closely aligns with perceptual realism.

3. Utilizing diagnostic consensus rates from ophthalmologists, we investigate the impact of data organization on the performance of generative models to reveal their impact from a technical and academic perspective.

## Materials and methods

### StyleGAN2-ADA: Style transfer-based generative model

Style transfer (ST) is a technique that combines the style of one image with the content of another to create a new image. This technique was first proposed by [29] and is now a crucial part of the generative model field. It introduces a new approach by reflecting the characteristics of the style image while preserving the basic structure of the content image, distinguishing it from traditional generative models.

However, this technique has several significant limitations. First, because ST processes each image independently, it is not easy to maintain consistency between sequence images. Second, while ST effectively modifies at modifying low-level style elements like texture, it needs to work on more complex high-level style transformations. Third, image generation using ST is

fully automated, making it challenging for users to control the impact on the result. This makes tasks such as emphasizing or offsetting certain styles or content complex for the user.

To overcome the limitations of ST, StyleGAN [30], which applies ST to the generator of PGGAN [31] was proposed. This model uses a mapping network to apply different styles to various scales for training. During this process, adaptive instance normalization (AdaIN) prevents changes in scale and variance as each layer passes and performs the input style role. This model design allows for more high-dimensional style transformations than ST and also allows user control.

Furthermore, StyleGAN2 [32], which applies path length regularization and removes artifacts during the image generation, enhanced the generation performance. For situations with limited data, StyleGAN2-ADA, which applies various forms of augmentation to the images provided to StyleGAN2's discriminator, was developed to learn the generative model effectively. In this paper, due to the need for an effective generative model in situations with limited data, StyleGAN2-ADA was used to generate fake strabismus images.

## Evaluation metric of generative models

### Metrics for evaluating image quality.

- Peak signal-to-noise ratio(PSNR) and structural similarity index(SSIM)
  PSNR is a metric used to evaluate image quality by measuring the ratio of signal to noise between the original and reconstructed images. A higher PSNR value indicates a lower loss in the reconstructed image. SSIM is a measure of image quality that compares the structural similarity between the original and reconstructed images. It considers luminance, contrast, and structural information to provide a value between -1 and 1, where a higher value indicates better similarity. SSIM and PSNR are widely used metrics in various fields related to image processing. However, both of these metrics are only intended to evaluate the quality of the images and have limitations in assessing the perceptual performance as perceived by humans. In other words, these metrics have limitations in confirming perceptuality.

### Metriccs to evaluate the perceptuality.
When researching generative models, various methods have been proposed to overcome the cognitive performance limitations of the two metrics mentioned above. IS and FID are the most commonly used metrics in this regard.

- Inception Score
  IS is a metric used to evaluate the quality and diversity of generated images, where higher values indicate better results. IS incorporates perceptuality by directly using a deep learning network for metric computation. When predicting class labels for generated samples, it measures the confidence of the inception network. This metric gives a good score when both fidelity ($p(y|image)$) distribution, which represents the dependency of predicted results on specific labels, and diversity ($p(y)$) distribution are high. From observation (1), it can be concluded that higher fidelity leads to better results. Conversely, a more uniform diversity distribution is considered better for generating results. As the two distributions diverge, it indicates better results from the perspective of the generative model, which means a higher KL Divergence ($D_{KL}$) value represents better results.

$$IS = \exp\left(E_{\text{image}}[D_{\text{KL}}(p(y|\text{image}) \| p(y))]\right) \tag{1}$$

- Fréchet Inception Distance

  FID [33] is a metric that calculates the distance between the feature representations of generated and authentic images. This metric was proposed based on the idea that if fake images are generated well by a generative model, the feature maps at the same layer of a deep learning model should be similar for real and generated images. Unlike IS, which evaluates generated images alone, FID compares actual and generated samples. Additionally, FID uses Fréchet distance to reflect perceptual differences, unlike PSNR, which measures pixel-level distances between two images. A lower FID value indicates a higher similarity between the actual and generated distributions. Thus, a lower FID value represents better alignment in terms of similarity between the actual and generated distributions. (2) represents the calculation of FID, where $\mu_{real}$ and $\mu_{gen}$ represent the mean feature representations of actual and generated images, respectively. In contrast $\Sigma_{real}$ and $\Sigma_{gen}$ represent the covariance matrices of actual and generated images, respectively.

$$\text{FID} = \|\mu_{\text{real}} - \mu_{\text{gen}}\|_2^2 + \text{Tr}(\Sigma_{\text{real}} + \Sigma_{\text{gen}} - 2(\Sigma_{\text{real}}\Sigma_{\text{gen}})^{1/2}) \tag{2}$$

The generative model research field has established some quantitative benchmarks thanks to the emergence of perceptual metrics based on deep learning, including IS and FID. In general, when analyzing the performance of a generative model, FID is most often used. In the case of IS, it was widely used before FID was introduced. However, its use has decreased. It is disadvantageous in measuring perceptuality than FID because it does not utilize actual data distribution. In this context, our study uses only FID for quantitative performance verification of generative models.

## Study design

This study was performed in accordance with the tenets of the Declaration of Helsinki and approved by the Institutional Review Board of Pusan National University Yangsan Hospital (IRB No. 04-2021-008, 05-2020-048), with waiver of informed consent granted for this retrospective study. The information of all participants were completely anonymized before the access of researchers. For the purposes of this research, the data was accessed by the researchers between May 4, 2020, and April 21, 2021.

## Datasets

The data used in this study were collected from the Ophthalmology Department of Pusan National University Yangsan Hospital. The collected data comprised 996 images of normal eyes and 1253 images of strabismic eyes. The images collected are of the patient's entire face. As the identity of the patients could be identified from the full-face images, there may be issues with personal information protection. Therefore, before delivering the data to the researchers, we cropped all collected data so that only both eye parts could be seen. We also conducted a separate data selection process to construct a high-quality dataset. The data selection criteria involved showing all collected images to 10 independent medical professionals and asking for diagnostic results for each data. When showing data to medical professionals, we randomly shuffled the order of the images during survey creation to prevent bias from sequence and continuity. After the survey, we composed the research dataset only with images that at least five medical professionals diagnosed the same. Ultimately, we constituted the dataset with 900 images of normal eyes and 896 images of strabismic eyes. We used 800 images from each class to train the deep-learning model. The validation dataset was not initially separated; when

**Table 1. Summary of dataset composition used in each experiment.** ODCR stands for Ophthalmologist Diagnosis Consensus Rate. For the test dataset, we selected 100 normal eye images and 96 strabismus eye images from the data collected initially, which had an ODCR of 80% or higher. The generated data was exclusively used for training and not used for validation or testing.

| Experiment | | | real | | | | | | generated | |
|---|---|---|---|---|---|---|---|---|---|---|
| | | | normal | | | strabismus | | | normal | strabismus |
| | | | training | validation | test | training | validation | test | training | training |
| Exp. 1 | | | 640 | 160 | 100 | 640 | 160 | 96 | - | - |
| Exp. 2 | for generative model (ODCR) | 60% | 611 | 153 | 100 | 629 | 157 | 96 | - | - |
| | | 70% | 556 | 139 | 100 | 550 | 138 | 96 | - | - |
| | | 80% | 480 | 120 | 100 | 455 | 114 | 96 | - | - |
| | for classifier(ODCR) | 60% | 640 | 160 | 100 | 640 | 160 | 96 | 3000 | 3000 |
| | | 70% | | | | | | | 3000 | 3000 |
| | | 80% | | | | | | | 3000 | 3000 |
| Exp. 3 | | | 640 | 160 | 100 | 640 | 160 | 96 | 500/1000/2000/3000 | 500/1000/2000/3000 |

starting the deep learning model training, we designed the experiment to select 20% of the training dataset randomly automatically. The remaining data (100 images of normal eyes and 96 images of strabismic eyes) was used as a test dataset. We also composed a separate dataset based on the previously conducted survey to check the impact of ODCR on the quantitative performance of the generative model. These datasets were each constructed based on ODCR values of 60% (864 normal images, 882 strabismic images), 70% (795 normal images, 784 strabismic images), and 80% (700 normal images, 665 strabismic images), respectively. All information regarding the datasets used in this research can be found in Table 1.

## Experimental setup

Our research aims to determine whether data augmentation techniques based on generative models can have an effect similar to that of actual data collection. To verify this, we conducted three experiments.

**Experiment 1. Verification of the effectiveness of traditional data augmentation techniques in situations with limited data.** Before verifying the effects of data augmentation based on generative models, we first experimented on whether traditional data augmentation techniques could affect performance improvement even when the data is sparse. We prepared a test group and a control group for this experiment based on the same dataset (900 normal and 896 strabismus images). In the case of the control group, we did not use data augmentation techniques on the training dataset. For the experimental group, we designed it to randomly perform horizontal flip and rotation before the training dataset goes into the model input. Except for the presence or absence of data augmentation, all settings necessary for learning were kept the same. In addition, we used accuracy and area under the receiver operating curve(AUC) for performance verification.

**Experiment 2: Verification of the relationship between expert selection statistics and the performance of the generative model.** We hypothesized a correlation between the consensus among expert diagnoses and the performance of our generative model. This is because a high ODCR suggests that the data is easier to perceive by human standards, and we reasoned that a deep learning model would be more capable of producing plausible data when trained on such perceptually rich input. To test this, we surveyed ten ophthalmology experts, asking them to diagnose all data samples we had. We then separated the data into distinct sets based on ODCR and trained a StyleGAN2-ADA model on each set.

We employed the FID, a widely accepted metric for the quantitative performance assessment of generative models, to evaluate the generative models. Besides the FID, we also trained a classifier(ResNet50) with the generated samples to evaluate their performance, as the FID, though a common metric, does not perfectly capture perceptuality. Training the classifier on the generated samples(3000 generated normal images and 3000 generated strabismic images) alongside the real ones was optimal to gauge whether the generated data could substitute for collected data. Except for the data set composition, all experiment conditions remained the same as in the previous experiment.

**Experiment 3: Assessing the feasibility of data augmentation based on generative models.** Finally, we experimented to verify whether data augmentation based on generative models could have the same effect as actual data collection. We collected samples of normal and strabismic eye images from the generator of a StyleGAN2-ADA model trained on data with 80% diagnostic agreement among experts. The samples varied: 500, 1000, 2000, and 3000 for each condition. We then trained the classifier using each of these sample groups in conjunction with the dataset from the first experiment. For the classifier, we used both ResNet50 and ResNext101 models.

This experiment applied traditional data augmentation techniques to the training dataset. Fine-tuning and early stop (patience 10) were also used to prevent an undertrained state in the initial stages. Other settings included using the RAdam optimizer, a learning rate of 0.0001, and batch size 32. Our system was implemented based on the PyTorch Ubuntu 20.04 operating system framework. The implemented system was trained and evaluated on a workstation with an AMD Ryzen 9 5950x 16-core CPU, 128GB RAM, and two NVIDIA RTX 3090 24GB GPUs.

For performance verification, we used accuracy and AUC. To verify the effectiveness of the proposed method in addressing overfitting, we quantified the differences between the training and validation accuracy and between the training and validation loss at the end of the learning process.

For this study, the experimental code conducted is available at the following link: https://github.com/joojaehan/stra-classfication-pnu.

## Results

In this paper, we conducted three experiments to examine the effectiveness of generative model-based data augmentation. The data information used in each experiment can be found in Table 1.

### Results of verifying the effectiveness of traditional data augmentation

In this experiment, we investigate whether traditional data augmentation techniques improve the deep learning model's prediction accuracy and overfitting problems, even in situations with highly sparse data. We used two deep learning-based classifiers (ResNet50 and ResNext101) to assess the performance. $L1$ and $L2$ in Figs 1 and 2 respectively represent experiments without data augmentation and experiments with data augmentation. From the results of the ResNet50 model in Fig 1, we observe that data augmentation leads to improvements in both accuracy (85.71% → 86.73%) and AUC (0.8598 → 0.8650). However, for the ResNext101 model, we noticed a decrease in accuracy (87.75% → 86.73%) and AUC (0.8769 → 0.8673) when using data augmentation. Nevertheless, examining Fig 2, we find that the reduction in the difference of loss values when using data augmentation indicates an improvement in the overfitting problem (ResNet50: 0.12 → 0.1, ResNext101: 0.075 → 0.049).

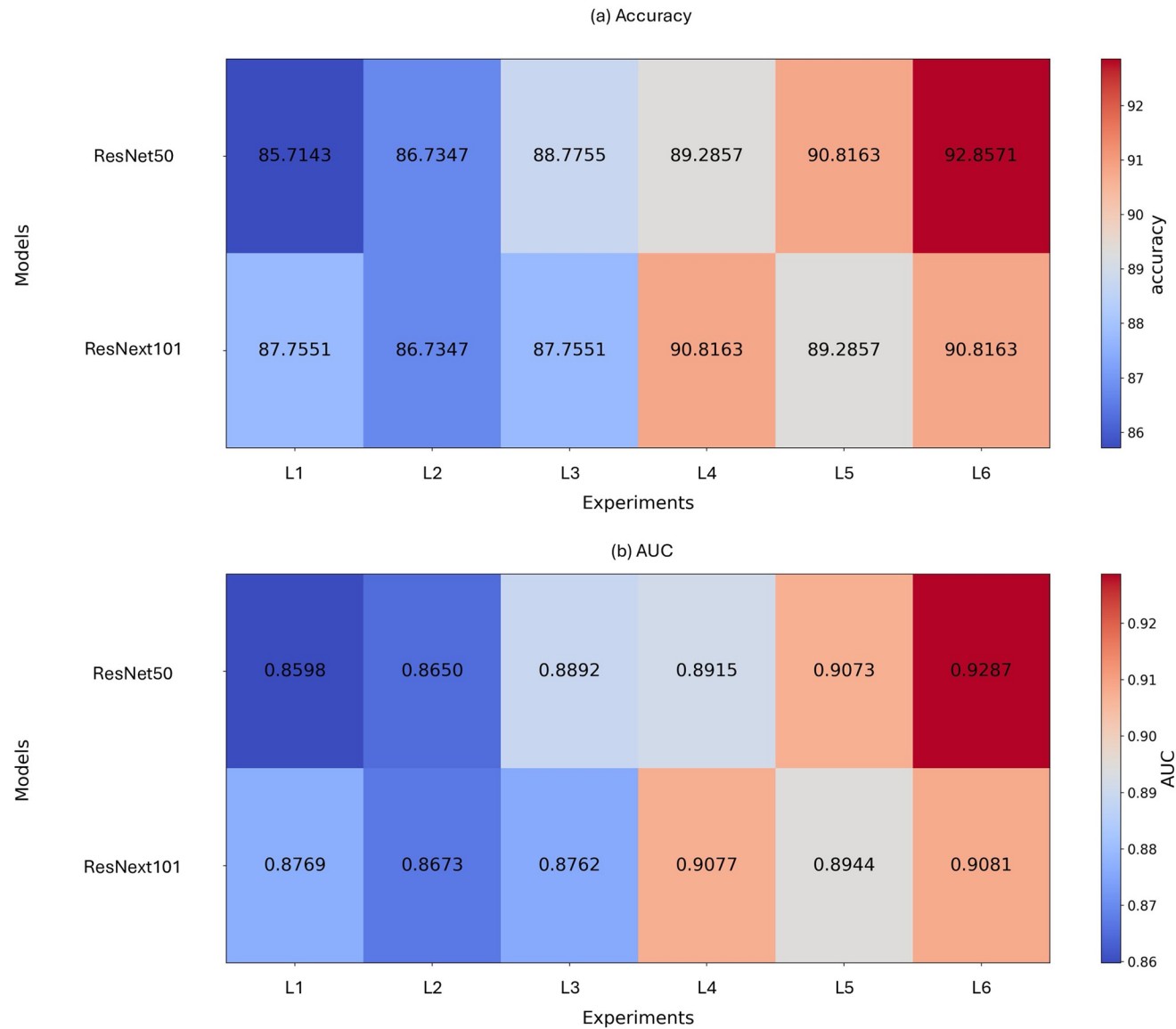

**Fig 1. Test results.** Test results (Accuracy(a) and AUC(b)) for 6 experiments. *L1* shows the results without data augmentation and generated data addition. *L2* represents the results with data augmentation only. *L3* to *L6* demonstrate the test results after adding 500, 1000, 2000, and 3000 generated data to the training dataset, followed by data augmentation.

### Generative model performance results according to ODCR

Through the first experiment, we confirmed that data augmentation improves the accuracy of deep learning models for automatic ocular diagnosis even in very scarce data and alleviates the overfitting problem. In this experiment, we validated the generation performance of the Style-GAN2-ADA, which will be used for data augmentation based on the generative model. This experiment aimed to verify the assumption that "generative models will produce better-quality fake data when trained with data that has a high agreement among ophthalmologists' diagnoses."

Experiment 2 area of Table 1 shows the dataset configurations used for these experiments, and Table 2 presents the results. Additionally, Fig 3 depicts samples of strabismic and normal

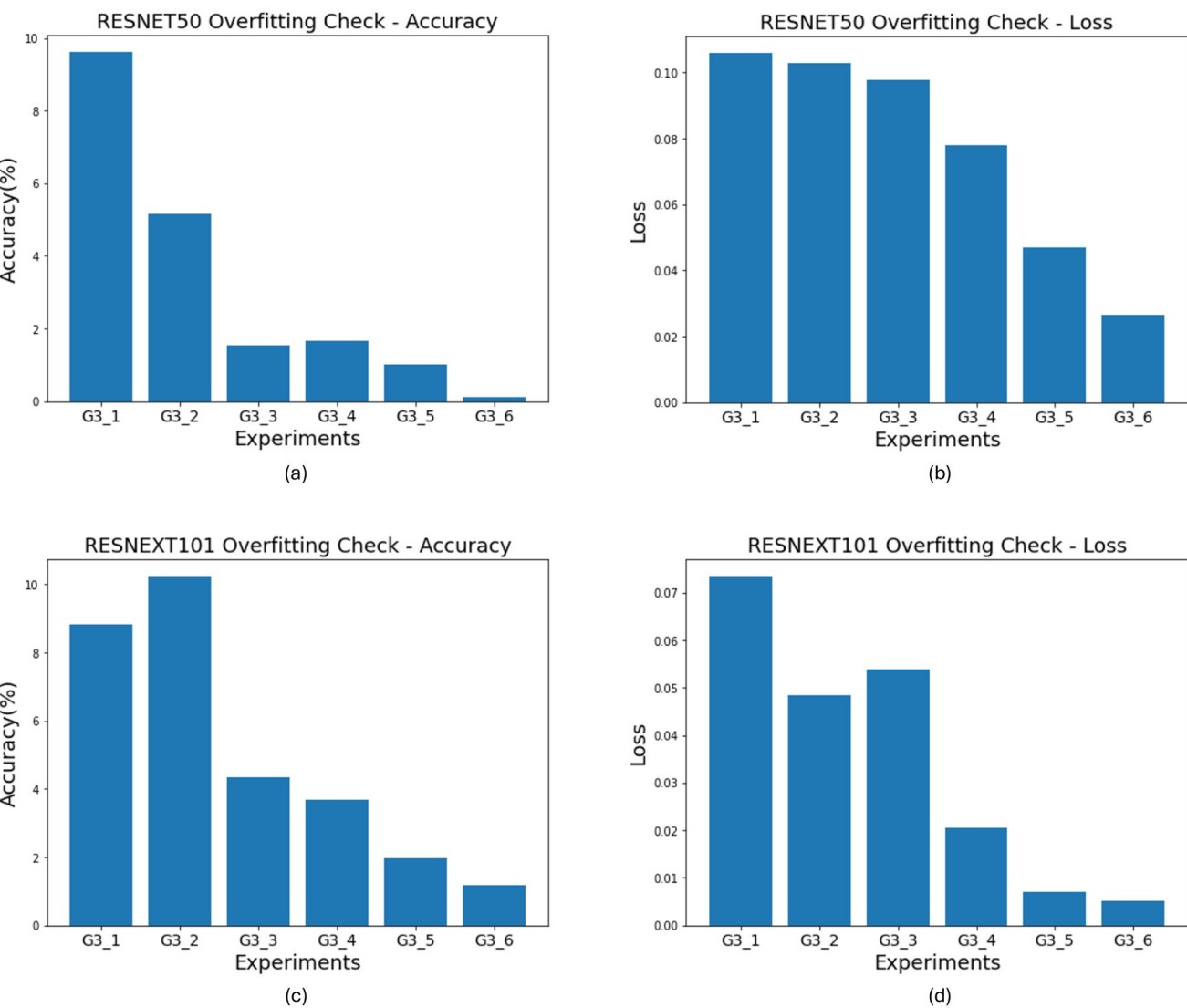

**Fig 2. Visualize overfitting problem: Difference between training and validation loss.** Visualization of the effectiveness in addressing overfitting problems when performing classification tasks based on the ResNet50 and ResNext101. Each experiment shows the difference in accuracy and loss between training and validation at the end of training. (a), (c) represents the accuracy difference, and (b), (d) represents the loss difference. (a) and (b) represent the results of the ResNet50 model, while (c) and (d) represent the results of the ResNext101 model.

**Table 2. Generation performance according to ODCR.**

| ODCR | Normal(FID) | Strabismus(FID) | Accuracy(%) |
|---|---|---|---|
| 60% | 9.4565 | 8.0054 | 88.36 |
| 70% | 9.5269 | 8.1885 | 89.43 |
| 80% | 11.6624 | 8.3361 | 92.85 |

Results of training and testing the classifier (ResNet50) using data generated by the Generator mapped to the ODCR values of the StyleGAN2-ADA model. The FID values for each ODCR condition were also evaluated during the experiment.

(a)

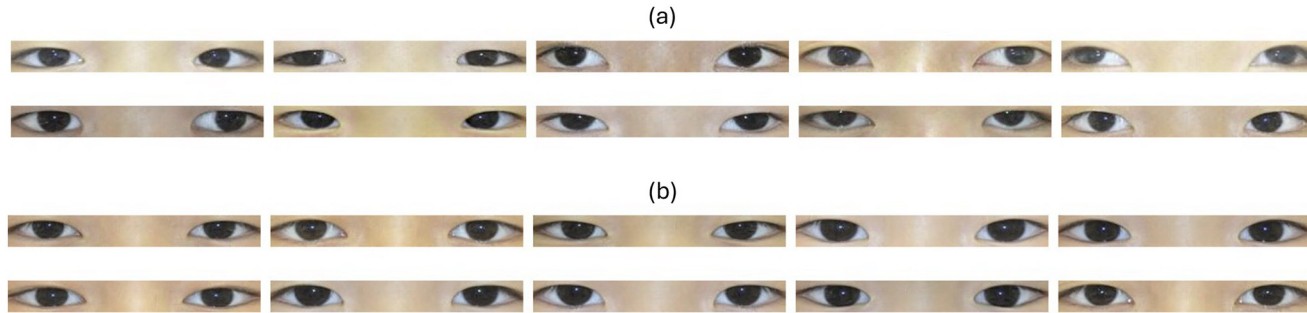

(b)

**Fig 3. Generated samples.** Data samples generated based on StyleGAN2-ADA. Both squint eye and normal eye samples were extracted from the Generator of the trained model, which was trained with an ODCR of 80% as the criterion. (a) represents the generated strabismus eye sample, and (b) represents the generated normal eye sample.

eyes generated based on a generator with an ODCR of 80%. Through Fig 3, we confirmed that the generated eyes samples are perceptually indistinguishable from real ones.

Subsequently, we analyzed the performance quantitatively. Regarding the FID, contrary to our expectations, for the diseased eyes, the FID showed almost no change even with an increase in ODCR, while for the normal eye, the FID results worsened. However, since FID and other quantitative metrics of generative models do not fully represent their performance, we conducted additional performance verification.

We generated fake data samples (3000 samples each for normal eye and strabismus) from each generator corresponding to different ODCR values and combined them with real training datasets. Then, we performed a classification task using a ResNet50 classifier. Surprisingly, as shown in Table 2, while ODCR seemed insignificant in terms of FID, it was found to influence the improvement of deep learning model performance in terms of accuracy (88.36% → 89.43% → 92.85%) as revealed by the classifier.

## Generative model-based data augmentation results

Finally, we examined the performance results when the data samples generated through the StyleGAN2-ADA model were included in the training dataset. Since the second experiment revealed that the quality of the generated samples was influenced by the ODCR, we used a generator trained with an ODCR of 80% for this experiment. Additionally, to observe the performance changes with different numbers of added generated samples, we varied the number of generated samples added to the training dataset.

$L3$ to $L6$ in Fig 1 represent situations where 1000, 2000, 4000, and 6000 additional generated samples (equally split between normal and diseased eyes) were included in the training dataset. From Fig 1, we observed that both ResNet50 and ResNext101 models showed improved performance when the generated samples were added compared to data augmentation alone. Notably, for ResNet50, adding 6000 samples resulted in an improvement of approximately 6% in accuracy and 0.0637 in AUC. Furthermore, as shown in Fig 2, we observed significant improvements in mitigating the overfitting problem(Loss difference between validation and training, ResNet50; $0.1027(L2) \rightarrow 0.0265(L6)$, ResNext101; $0.0483(L2) \rightarrow 0.0051(L6)$).

To present the qualitative results of this study, we prepared Fig 4. It showed the results when ResNet50 was trained and tested with 6000 additional generated samples. These images are all generated samples from the StyleGAN2-ADA, and we used GradCAM to visualize

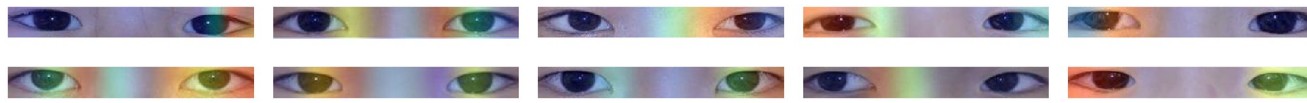

**Fig 4. GradCAM.** Visualization of the activation regions using GradCAM for the classifier(ResNet50).

which regions were activated by the classifier for each test sample. From the images, we can observe that the classifier focuses on the inner parts of both eyes for classification.

## Conclusion

Various studies have been conducted for automated strabismus diagnosis, and recently, methods based on deep learning have also been proposed. Primarily for machine learning-based automated diagnostic systems, deep learning, data collection, and processing play crucial roles in designing the entire system. Ideally, if the model can be generalized and a large amount of data can be collected, it is possible to design a high-performing model. However, in the medical field, there are limitations to data collection due to various factors such as privacy protection and disease prevalence.

While previous research has proposed methods that perform well using a substantial amount of data, there has yet to be a study that suggests improvements and analyzes them in situations where data is severely limited. To overcome this challenge, our study proposes a generative model-based data augmentation technique to address the data collection limitations in designing a deep learning-based automated glaucoma diagnosis system. We quantitatively and qualitatively assess the feasibility of this approach.

To design our proposed system, we implemented a generative data augmentation technique based on the StyleGAN2-ADA model. We evaluated the performance using two CNN-based models for classification. We conducted a series of three experiments to demonstrate the effectiveness of our approach. In the first experiment, we examined whether traditional data augmentation techniques are effective in situations with highly sparse data. While there was a modest quantitative improvement (accuracy, AUC), we observed significant benefits in addressing overfitting issues. In the second experiment, we investigated the relationship between ODCR and generation performance during the training of the generative model. Although there were marginal improvements from the perspective of FID, we found that higher ODCR values led to enhanced performance when using the generated data to train classifiers. Finally, we validated our approach by varying the number of data samples generated by a generator trained on data with an ODCR of 80%. Through this experiment, we demonstrated that our generative model-based data augmentation outperforms traditional data augmentation, showing remarkable performance improvements and significant mitigation of overfitting issues.

In conclusion, our proposed method effectively leverages data composition and generative models to improve performance while mitigating overfitting, even in scenarios with severely limited data. Currently, our proposed data generation method cannot finely control the generated samples. In future research, our goal is to design a generative model that allows for precise adjustment of various features such as pupil position, age, and iris color.

## Author Contributions

**Conceptualization:** Jaehan Joo, Donghwan Kim, Su-Jin Kim, Suk Chan Kim.

**Data curation:** Jaehan Joo, Sang Yoon Kim, Ji-Eun Lee, Seung Min Lee, Su Youn Suh, Su-Jin Kim.

**Formal analysis:** Jaehan Joo, Suk Chan Kim.

**Methodology:** Jaehan Joo, Sang Yoon Kim, Donghwan Kim, Suk Chan Kim.

**Software:** Donghwan Kim.

**Supervision:** Suk Chan Kim.

**Validation:** Jaehan Joo, Sang Yoon Kim, Ji-Eun Lee, Seung Min Lee, Su Youn Suh, Su-Jin Kim, Suk Chan Kim.

**Visualization:** Jaehan Joo.

**Writing – original draft:** Jaehan Joo.

**Writing – review & editing:** Su-Jin Kim, Suk Chan Kim.

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
