## [Decision Letter · Decision Letter 0]

6 Feb 2024

PONE-D-23-27056Enhancing Automated Strabismus Classification with Limited Data: Data Augmentation Using StyleGAN2-ADAPLOS ONE

Dear Dr. Kim,

Thank you for submitting your manuscript to PLOS ONE. After careful consideration, we feel that it has merit but does not fully meet PLOS ONE’s publication criteria as it currently stands. Therefore, we invite you to submit a revised version of the manuscript that addresses the points raised during the review process.

We look forward to receiving your revised manuscript.

Kind regards,

Neelam Pawar

Academic Editor

PLOS ONE

Journal Requirements:

   "This work was supported by 2022 BK21 FOUR Program of Pusan National University. This study was supported by Research Institute for Convergence of biomedical science and technology (30-2021-000), Pusan National University Yangsan Hospital"

   "NO, The funders had no role in study design, data collection and analysis, decision to publish, or preparation of the manuscript."

5. Please note that PLOS ONE has specific guidelines on code sharing for submissions in which author-generated code underpins the findings in the manuscript. In these cases, all author-generated code must be made available without restrictions upon publication of the work. Please review our guidelines at https://journals.plos.org/plosone/s/materials-and-software-sharing#loc-sharing-code and ensure that your code is shared in a way that follows best practice and facilitates reproducibility and reuse.

Additional Editor Comments:

After revision paper can be considered for publication.

Reviewers' comments:

Reviewer's Responses to Questions

**Comments to the Author**

1. Is the manuscript technically sound, and do the data support the conclusions?

Reviewer #1: Yes

2. Has the statistical analysis been performed appropriately and rigorously? 

Reviewer #1: Yes

3. Have the authors made all data underlying the findings in their manuscript fully available?

Reviewer #1: Yes

4. Is the manuscript presented in an intelligible fashion and written in standard English?

Reviewer #1: Yes

5. Review Comments to the Author

Reviewer #1: In this paper, the authors are proposed “Enhancing Automated Strabismus Classification with Limited Data: Data Augmentation Using StyleGAN2-ADA

" .

The strengths of the paper are that it is well structured, the description of the related work is well done and that results are extensively compared to results of the similar research.

1. Authors should draw a graphical abstract of this work.

2. Explain the novelty of the proposed approach.

3. Proofread the entire manuscript once again.

6. PLOS authors have the option to publish the peer review history of their article (what does this mean?). If published, this will include your full peer review and any attached files.

Reviewer #1: No

---

## [Author Response · Author response to Decision Letter 0]

5 Mar 2024

1. Authors should draw a graphical abstract of this work.

- Response

We have taken your comment into consideration and have created a Graphical Abstract for our paper. It has been attached as a separate document titled “Graphical Abstract” upon resubmission. We are grateful for your comments, as we believe they (the comments) have helped to enhance the understanding of our readers and have also contributed to the improvement of the quality of our paper.

Thank you for your valuable feedback.

2. Explain the novelty of the proposed approach

- Response

In this paper, we propose a method that leverages generative models to enhance the performance of deep learning-based classification models under extremely limited conditions. While this approach has been utilized in other fields, our research introduces this method for the first time in the area of strabismus classification, marking a novel contribution. Specifically, instead of using generic generative models, we demonstrate the capability to create fake data resembling real images through fine-tuning based on the StyleGAN2-ADA model, which maximizes perceptual realism.

Beyond the technical novelty, our research adds significant originality by analyzing the performance of the model based on the diagnostic consensus of ophthalmologists (referred to as ODCR in the paper) during the validation phase. This highlights the impact of high-quality data composition on performance in medical data based research, not just model construction. These details are thoroughly described in the sections "Proposed Method and Contributions" and "Conclusion" of the paper.

To summarize the novelty of our paper:

a. We are the first to conduct performance improvement research in the field of strabismus classification using generative models.

b. We successfully implemented a high-quality generative model to create data that closely resembles real images.

c. We emphasized the importance of high-quality data through performance analysis based on the diagnostic agreement rate (ODCR) of medical experts.

3. Proofread the entire manuscript once again

- Response

We have proofread every section of our paper and addressed the necessary revisions. These adjustments are documented in the marked version of the amended manuscript. The issues we considered when performing proofreading are as follows:

a. Converting sentences from passive voice to active voice

b. Correcting grammatically incorrect sentences

c. Modifying symbols or formats that do not conform to the journal’s guidelines

d. Corrected typos in author's name and affiliations

e. Revising sentences that could be misunderstood

---

## [Decision Letter · Decision Letter 1]

24 Apr 2024

Enhancing Automated Strabismus Classification with Limited Data: Data Augmentation Using StyleGAN2-ADA

PONE-D-23-27056R1

Dear Dr. Kim,

We’re pleased to inform you that your manuscript has been judged scientifically suitable for publication and will be formally accepted for publication once it meets all outstanding technical requirements.

Kind regards,

Oliver Giudice, Ph.D.

Academic Editor

PLOS ONE

Additional Editor Comments (optional):

The authors addressed all the minor issues raised by the reviewers. The paper is now ready for publication.

Reviewers' comments:

Reviewer's Responses to Questions

**Comments to the Author**

1. If the authors have adequately addressed your comments raised in a previous round of review and you feel that this manuscript is now acceptable for publication, you may indicate that here to bypass the “Comments to the Author” section, enter your conflict of interest statement in the “Confidential to Editor” section, and submit your "Accept" recommendation.

Reviewer #1: All comments have been addressed

2. Is the manuscript technically sound, and do the data support the conclusions?

Reviewer #1: Yes

3. Has the statistical analysis been performed appropriately and rigorously? 

Reviewer #1: Yes

4. Have the authors made all data underlying the findings in their manuscript fully available?

Reviewer #1: No

5. Is the manuscript presented in an intelligible fashion and written in standard English?

Reviewer #1: Yes

6. Review Comments to the Author

Reviewer #1: The authors proposed "Enhancing Automated Strabismus Classification with Limited Data: Data Augmentation Using StyleGAN2-ADA"

1. proofread the entire manuscript

All the comments are addressed by authors.

7. PLOS authors have the option to publish the peer review history of their article (what does this mean?). If published, this will include your full peer review and any attached files.

Reviewer #1: No

---

## [Editor Report · Acceptance letter]

14 May 2024

PONE-D-23-27056R1 

PLOS ONE

Dear Dr. Kim, 

I'm pleased to inform you that your manuscript has been deemed suitable for publication in PLOS ONE. Congratulations! Your manuscript is now being handed over to our production team.

Kind regards, 

on behalf of

Dr. Oliver Giudice 

Academic Editor

PLOS ONE